# The Development of Prenatal Muscle Satellite Cells (MuSCs) and Their Epigenetic Modifications During Skeletal Muscle Development in Yak Fetus

**DOI:** 10.3390/biology13121091

**Published:** 2024-12-23

**Authors:** Guoxiong Nan, Wei Peng, Shangrong Xu, Guowen Wang, Jun Zhang

**Affiliations:** College of Animal Husbandry and Veterinary Science, Qinghai University, Xining 810016, China; ngx196414@163.com (G.N.); 2009990033@qhu.edu.cn (W.P.); 2008.xushangrong@163.com (S.X.)

**Keywords:** yak, skeletal muscle, muscle satellite cell, epigenetic modification

## Abstract

Yaks, which are ruminant animals primarily distributed in the Qinghai–Tibet Plateau, have an unclear skeletal muscle development process. This paper investigates the changes of muscle satellite cells and their epigenetic modifications in yak skeletal muscles during the developmental process. These data provide a theoretical basis for the growth and development of yak skeletal muscles.

## 1. Introduction

Yaks (*Bos grunniens*) are ruminant animals adapted to high-altitude, low-oxygen environments, thriving at elevations between 3000 and 5000 m. Predominantly found across the Tibetan Plateau and its surrounding areas. Yaks play a crucial role in the ecological stability, economic development, and cultural heritage of the region. Their adaptation to extreme environments is attributed to their unique physiological and genetic characteristics [1]. Yak meat, a significant byproduct, serves as a vital food source for local residents and holds high economic value due to its quality and nutritional benefits, positioning it as a natural, green alternative in the global beef market [2].

Skeletal muscle, the predominant muscle tissue in yaks, significantly influences survival, reproduction, and meat production. Comprising 30–40% of body weight, skeletal muscle is composed of muscle fibers formed through the fusion of myogenic progenitor cells [3]. Its growth and development involve a complex biological process of cell proliferation, differentiation, fusion, and maturation [4]. Early in embryonic development, precursor cells from the notochord and mesoderm differentiate into myoblasts. These myoblasts, influenced by signaling pathways such as Wnt, Hedgehog, and FGF, form primary myotubes [5]. Primary myotubes, multinucleated structures resulting from the fusion of multiple myoblasts, serve as precursors to skeletal muscle fibers. During this process, some myoblasts remain unfused around the muscle fibers and become satellite cells [6]. As development progresses, primary myotubes mature, while myoblasts continue to proliferate and differentiate, forming secondary myotubes in addition to primary ones [7]. Secondary myotubes primarily arise from unfused myoblasts of the primary myotubes, and their formation significantly exceeds that of primary myotubes during later embryonic stages [8]. Secondary myotubes further develop into mature skeletal muscle fibers, thickening and maturing the sarcoplasmic reticulum, evenly distributing nuclei around the muscle fibers, and forming well-aligned sarcomeres visible under a light microscope [9]. The process of skeletal muscle development is sequential, starting with the formation of primary myotubes and progressing to secondary myotubes, which form in greater numbers as development advances. After birth, the number of muscle fibers remains relatively constant, with only maturation and hypertrophy occurring [10]. Thus, fetal skeletal muscle development is crucial for muscle quality postnatally. However, research on yak fetal skeletal muscle development remains limited.

In skeletal muscle differentiation and development, myogenic progenitor cells are essential prenatally, while satellite cells are crucial postnatally. Both cell types exhibit stem cell characteristics, including the ability to proliferate, fuse, and form new muscle fibers, and both express PAX7 [11]. Among paired box transcription factors, PAX3 and PAX7 are pivotal in skeletal muscle development, with PAX3 and PAX7 expressed in myogenic progenitor cells and PAX7 present in all MuSCs [12]. Muscle satellite cells (MuSCs), located between the muscle fiber basal lamina and the muscle membrane, are critical muscle stem cells involved in muscle growth, repair, and regeneration [13]. However, current research on PAX7-expressing satellite cells primarily focuses on postnatal regeneration and repair, with limited studies on their role during embryonic development [14].

Epigenetic modifications, including DNA methylation and histone modification, play a significant role in the proliferation, differentiation, and functional development of muscle cells [15]. Recent studies underscore the importance of epigenetic regulation in muscle cell function and characteristics [16]. However, despite the growing interest in epigenetics, little is known about how DNA methylation and histone modification specifically influence yak fetal muscle development, particularly in the context of satellite cells. Therefore, investigating the development of satellite cells marked by PAX7 and their associated epigenetic modifications during embryogenesis is crucial for elucidating yak skeletal muscle growth and development.

This research aims to explore the changes in satellite cells and their associated epigenetic modifications during yak fetal development, with a particular focus on DNA methylation and histone modifications. The findings may provide new insights into yak skeletal muscle growth and development, which could inform strategies to enhance production performance in the future.

## 2. Materials and Methods

### 2.1. Animals

All animal experimentation was performed in accordance with institutional ethical requirements and were approved by the Qinghai University Animal Ethics Committee. Yak fetuses of various lengths were collected from Qinghai Haiyan Xiahua Food Co., Ltd., (Haibei, China). At each time point, samples from three yaks were harvested as biological replicates, and the body weights and lengths of yaks during each embryonic stage were recorded. The specimens were cleaned with 75% ethanol, immediately placed in sealed bags, and transported at 4 °C to the laboratory. Immediately, body measurements were recorded, and the specimens were dissected to obtain the longissimus dorsi muscle, which was then preserved in 4% paraformaldehyde. Some samples were frozen immediately in liquid nitrogen and stored there until further use.

### 2.2. Hematoxylin–Eosin Staining

Muscle tissue fixed in 4% paraformaldehyde was cut into appropriately sized blocks, followed by dehydration in a gradient of alcohol. After clearing with xylene, the tissue was embedded in paraffin overnight. The embedded blocks were sectioned into 4 μm thin slices using a microtome. The sections were floated on warm water, mounted on slides, and then dried to produce paraffin sections. The prepared paraffin sections were deparaffinized using xylene and a gradient of alcohol, stained with hematoxylin and eosin, and then mounted using a mounting medium composed of neutral gum and xylene.

### 2.3. Immunofluorescence Staining and Quantification of Positive Cells

The paraffin sections were deparaffinized with xylene and a gradient of alcohol. Antigen retrieval was performed with sodium citrate for 20 min. Afterward, the sections were permeabilized with 0.3% Triton X-100 in PBS for 10 min at room temperature. The sections were then blocked with 10% donkey serum for 1 h. Primary antibodies were incubated overnight at 4 °C, and fluorescently labeled secondary antibodies were applied at room temperature for 60 min. Nuclei were counterstained with 1 μg/mL DAPI, and the slides were mounted with an anti-fade reagent. Images were captured using an inverted fluorescence microscope. All image processing and analyses were performed using ImageJ (1.54k). Primary antibodies were as follows: rabbit anti-PAX7 (1:200, Abcam, #ab187339, Cambridge, UK); mouse anti-Myosin hc (1:300, R&D Systems, #MAB4470, Minneapolis, MN, USA); mouse anti-PCNA Monoclonal (1:200, Immunoway, #YM3031, Plano, TX, USA); mouse anti-5-Hydroxymethylcytosine (1:200, Active Motif, Cat#39999, Carlsbad, CA, USA); mouse anti-Histone H3K4me3 (1:200, Active Motif, Cat#91263); mouse anti-5-Methylcytosine (1:200, Active Motif, Cat#39649). All Alexa Fluor secondary antibodies against appropriate species were raised in donkey and used at 1:500 dilution (Thermo Fisher Scientific, Waltham, MA, USA). To quantify the positive cells in the cross-section of the longest dorsal muscle at each gestational age, we randomly selected three cross-sections of the longest dorsal muscle from individuals of the same developmental length. The antibodies involved in this experiment have good specificity in yaks (Appendix A).

### 2.4. Statistics

To statistically analyze the significance of differences between two or multiple groups, the student’s *t*-test or one-way analysis of variance (ANOVA) followed by the least significant difference (LSD) method was employed, respectively. All quantitative data were demonstrated as the mean ± standard error of the mean (SEM), and differences were defined as statistically significant at *p* < 0.05.

## 3. Results

### 3.1. Morphology of the Longissimus Dorsi Muscle at Different Developmental Stages in Yak Fetus

#### 3.1.1. Morphometric Data of the Fetal Yak at Different Developmental Stages

Yak fetuses at various developmental stages were collected, and crown–rump lengths (CRLs) were measured (Table 1). Images of the fetuses are shown (Figure 1). The fetal age was estimated based on these measurements. Variations in CRL can occur due to breed differences, rearing conditions, and post-mortem changes, especially in the later stages of pregnancy. Therefore, using the second edition of “*Veterinary Obstetrics*”, practical experience, and data on CRL and gestational age (GA), the approximate ages of the collected yak fetuses were estimated (Table 1). CRL is an important parameter that is closely associated with GA and developmental stage. Generally, as GA progresses, the crown–rump length increases. In the early stages of pregnancy, specific crown–rump length values correspond to particular developmental milestones. As the weeks pass, the crown–rump length lengthens, and the embryo develops more complex organs and systems. The developmental stage can be estimated based on the crown–rump length and GA, with each stage having characteristic morphological and physiological features. The smallest sample measured 17 mm, indicating that fetal cells were just beginning to form but were not fully developed. The GA of this period was judged as 37 days. At a length of 70 mm, the fetus displayed typical bovine features with the body wall closed and the abdominal wall prominent, suggesting approximately 2 months of gestation. At 100 mm in length, male fetuses had developed scrota, indicating an estimated age of around 11 weeks. At 270 mm, male fetuses had defined scrota, but no body or facial hair was present, suggesting a GA of less than 5 months. The largest sample, measuring 610 mm, had full body hair and nearly complete organ development, indicating an estimated age of 8 months.

Combining practical experience, CRL data, and fetal calf development to estimate the fetal age of the collected fetal calves, with 25 fetal calves distributed across different fetal ages. CRL, crown–rump length; GA, gestational age.

#### 3.1.2. Morphology of the Longissimus Dorsi Muscle at Different Developmental Stages

To determine the development process of yak skeletal muscle during the fetal period, H&E staining was performed on longissimus dorsi muscle tissues from fetuses at various stages (Figure 2). At an embryonic length of 17 mm, no muscle fibers were observed, indicating that this stage represents the early phase of muscle development. At 21 mm, primary muscle fibers began to appear, though in limited numbers, suggesting that muscle cells are starting to differentiate into primary fibers under specific signaling regulation. By 37 mm, numerous muscle fibers were present, and at 80 mm, fiber differentiation continued with an increasing number of muscle fibers. At 102 mm, the proliferation of primary muscle fibers peaked. At 122 mm, secondary muscle fibers began to appear. By 160 mm, a large number of secondary muscle fibers were evident, and by 165 mm, these fibers had grown to a size comparable to the primary muscle fibers. As development continued, by 240 mm, muscle fibers started to hypertrophy and mature, with fibers becoming thicker, interstitial tissue decreasing, and fiber arrangement becoming more organized. At 610 mm, just before birth, muscle fibers were neatly arranged into muscle bundles.

### 3.2. Localization of Muscle Satellite Cell and Muscle Fiber at Different Developmental Stages

In mammals, skeletal muscle expresses 12 different Myosin Heavy Chain (MyHC) genes with varying kinetics, making MyHC a specific marker for muscle fibers. PAX7 is a specific marker for MuSCs. To study the dynamic changes of satellite cells and muscle fibers, immunofluorescence staining for PAX7 and MyHC was performed on longissimus dorsi muscle tissues at different developmental stages (Figure 3a). At an embryonic length of 20 mm, MyHC was widely expressed in the longissimus dorsi muscle tissue, with only a few MyHC expressions forming ring-like structures. By 37 mm, most MyHC expressions were ring-shaped but incomplete. At 82 mm, MyHC expressions were predominantly ring-shaped, and by 102 mm, they were all complete rings, indicating that primary muscle fibers were largely formed. At 160 mm, secondary fibers appeared, with MyHC still showing ring-like expression. By 270 mm, muscle fibers had hypertrophied, and MyHC expression no longer exhibited a ring shape. By 610 mm, muscle fibers were neatly arranged. The immunofluorescence findings on muscle fiber development were consistent with the morphological results. In addition, we found that PAX7^+^ cells appeared in the middle of the myofiber cross-section at 82 mm and continued to do so at 160 mm, but most of them were present close to the myotube. Subsequently, we quantified the PAX7^+^ cells in each captured image. The results of the analysis showed that PAX7^+^ cells showed an overall upward trend, but decreased at 160 mm and 610 mm. Notably, PAX7^+^ cells spike at 160 mm, peaking in the number of positive cells at 270 mm (Figure 3b).

### 3.3. Proliferation of MuSCs in the Longissimus Dorsi Muscle at Different Developmental Stages of Yak Fetus

Proliferating Cell Nuclear Antigen (PCNA) is a reliable marker for assessing cell proliferation status. PAX7 is a specific marker for MuSCs. To examine the dynamic changes in satellite cell proliferation, immunofluorescence staining for PAX7 and PCNA was performed on longissimus dorsi muscle tissues. Immunofluorescence staining showed that all PAX7^+^ cells were able to co-localize with PCNA until the fetal bovine reached 82 mm in length (Figure 4a). Subsequently, we analyzed the ratio of PAX7^+^PCNA^+^ cells and PAX7^+^ cells in per image. The results of the analysis showed that the proliferation of yak muscle satellite cells gradually decreased until birth. It is worth noting that the ratio of PAX7^+^ PCNA^+^ cells to PAX7^+^ cells showed a decreasing trend, and the PAX7^+^ cells were basically in a proliferative state before the development to 82 mm, and the downward trend was more significant when the development reached 98 mm, and the PAX7^+^ cells basically stopped proliferating when the development reached 610 mm, and all entered the quiescent phase (Figure 4b).

### 3.4. DNA Methylation Modifications in MuSCs at Different Developmental Stages in the Longissimus Dorsi Muscle

#### 3.4.1. DNA 5mC Methylation Modification in MuSCs at Different Developmental Stages

DNA methylation, particularly the modification of 5-methylcytosine (5mC), is a widely studied epigenetic mechanism that acts as a stable repressive regulator of gene expression. To examine the dynamic changes in 5mC modification in MuSCs, we performed PAX7 and 5mC co-immunofluorescence staining on longissimus dorsi muscle of fetal cattle with different developmental lengths. Immunofluorescence staining showed that some PAX7^+^ cells expressed 5mC at different developmental lengths, while some 5 mC^+^ cells did not express PAX7 (Figure 5a). Subsequently, we analyzed the ratio of PAX7^+^5mC^+^ cells and PAX7^+^ cells in the longissimus dorsi cross-section of fetal cattle per developmental length. The analysis results showed that the ratio of PAX7^+^5mC^+^ cells and PAX7^+^ cells decreased first and then increased. Notably, the ratio of PAX7^+^5mC^+^ cells to PAX7^+^ cells decreases dramatically after development to 98 mm, and gradually increases after development to 120 mm, when the ratio of PAX7^+^5mC^+^ cells to PAX7^+^ cells gradually increases (Figure 5b).

#### 3.4.2. Expression of Demethylation Modification 5hmC in MuSCs at Different Developmental Stages

DNA hydroxymethylation is a key epigenetic regulation mechanism, involving the oxidation of 5-mC to 5-hydroxymethylcytosine (5-hmC) by TET protein family members, thereby facilitating the demethylation of DNA cytosines. To investigate the dynamic changes in 5hmC expression in MuSCs, immunofluorescence staining for PAX7 and 5hmC was conducted on longissimus dorsi muscle tissues from fetus at various developmental stages. Immunofluorescence staining showed that some PAX7^+^ cells expressed 5hmC at different developmental lengths, while some 5hmC^+^ cells did not express PAX7 (Figure 5c). We analyzed the ratio of PAX7^+^5hmC^+^ cells and PAX7^+^ cells in the longissimus dorsi muscle cross-section of fetal cattle per developmental length. The analysis results showed that the ratio of PAX7^+^5hmC^+^ cells and PAX7^+^ cells showed a dynamic trend. Developmental length of 21 mm, 98 mm, and 205 mm are the three turning points, respectively, and the change is the most critical before and after the developmental length of 98 mm (Figure 5d).

### 3.5. H3K4me3 Modification in MuSCs at Different Developmental Stages in the Longissimus Dorsi Muscle

Histone H3 lysine 4 trimethylation (H3K4me3) is one of the most extensively studied histone post-translational modifications and serves as a marker for transcriptional activation. To investigate the dynamic changes in H3K4me3 expression in yak MuSCs, immunofluorescence staining showed that almost all PAX7^+^ cells expressed H3K4me3, and only a very small number of PAX7^+^/H3K4me3^−^ cells appeared at 161 mm (Figure 6a). We analyzed the ratio of PAX7^+^H3K4me3^+^ cells and PAX7^+^ cells in the longissimus dorsi muscle cross-section of fetal cattle per developmental length. The results of the analysis showed that the overall trend of the ratio of PAX7^+^H3K4me3^+^ cells to PAX7^+^ cells did not change much, and the ratio was close to 1 (Figure 6b), indicating that the modification of histone H3K4me3 is critical for muscle satellite cells during fetal longissimus dorsi development.

## 4. Discussion

The development of skeletal muscle during the embryonic period determines muscle growth, and a series of complex processes during this phase lead to the formation of primary and secondary muscle fibers [17]. In this study, we analyzed the longissimus dorsi muscle tissue of yaks at different developmental lengths, combining the localization study of the muscle fiber-specific biomarker MyHC in the longissimus dorsi muscle tissue at various developmental stages of yak fetuses. It was preliminarily determined that primary skeletal muscle fibers in yaks begin to form at an embryonic length of 21 mm, are completely formed by 102 mm, and that secondary muscle fibers are essentially formed by 160 mm. Based on yak fetal age, primary muscle fibers start to form around 40 days post conception, are completely formed by approximately 11 weeks, and secondary muscle fibers are basically formed around 105 days. Subsequently, muscle fibers begin to differentiate and mature, forming complete muscle fiber bundles that are organized in a regular arrangement. Research by Picard B [18] found that primary muscle fibers are fully formed by 60 days post conception in beef cattle, after which primary muscle fibers start to differentiate, with secondary muscle fibers forming by 110 days and differentiation of secondary muscle fibers concluding by 180 days, at which point the total fiber count reaches its peak, followed by differentiation and maturation hypertrophy of fibers. These differences in muscle fiber development timing may contribute to the unique texture and quality of yak meat, which is known for its distinctive characteristics compared to beef. Doyle JL [19] found that genomic regions associated with muscle development vary among five different beef cattle breeds, and other scholars have noted that varying altitudes can lead to differences in protein phosphorylation levels in the longissimus dorsi muscle of yaks. The yaks used in this experiment were predominantly from the region, which exhibit significant differences in muscle growth performance compared to ordinary beef cattle [20].

MuSCs exhibit an asymmetric distribution within muscle tissue, with some accumulating at both ends of muscle fibers, while others are located in the central region of the fibers [21]. In this study, we combined the localization of the muscle satellite cell biomarker PAX7 and the muscle fiber-specific biomarker MyHC at different developmental stages of longissimus dorsi. Our findings indicate that most of the yak MuSCs are situated between the muscle fiber membrane and the basement membrane during fetal development, with only a small number found within the myotubes. Kuang S [21] discovered that PAX7^+^/Myf5^−^ satellite cells generate PAX7^+^/Myf5^+^ satellite cells through apex-basal symmetric division. This division asymmetrically produces basal PAX7^+^/Myf5^−^ satellite cells and apex PAX7^+^/Myf5^+^ satellite cells. Among these, PAX7^+^/Myf5^+^ satellite cells exhibit precocious differentiation behavior, while PAX7^+^/Myf5^−^ satellite cells make significant contributions to the satellite cell reservoir, with the latter being the majority. In conjunction with a comprehensive review by Guilhot C [22], much research has found that MuSCs undergo proliferation and differentiation. Most of the proliferating satellite cells are stored within the stem cell pool, i.e., between the muscle fiber membrane and the basement membrane, for muscle fiber repair and regeneration, while a small portion differentiate to form new muscle fibers. Our results support Kuang’s findings on the asymmetrical division of satellite cells and suggest that the differentiation of PAX7^+^/Myf5^+^ satellite cells at critical developmental stages plays a key role in the formation of new muscle fibers, where in the fetal calf lengths of 82 mm and 160 mm correspond to approximately 10 weeks and 105 days of yak gestation, respectively, aligning with the stages of primary and secondary muscle fiber formation. This suggests that during these two periods, MuSCs are predominantly apex PAX7^+^/Myf5^+^ satellite cells, contributing to the differentiation of new muscle fibers, while those located between the muscle fiber membrane and the basement membrane are likely PAX7^+^/Myf5^−^ satellite cells, serving to replenish the satellite cell stem cell pool. MuSCs exhibit heterogeneity [21]. Subsequently, Rocheteau P [23] found that high levels of PAX7 are associated with deeply quiescent MuSCs, which enter the cell cycle at a slower rate and possess lower metabolic activity, demonstrating enhanced adaptability during continuous transplantation. Once these cells cease proliferation, their metabolic rates slow down, and metabolic activity decreases. Combined with our research, it shows that when the yak fetal cow develops to 82 mm, the proliferation of PAX7^+^ cells slows down, and the cells begin to move and reserve to the stem cell pool, and when the yak is about to be born, the PAX7^+^ cells basically no longer proliferate, and the muscle satellite cells will all be stored in the stem cell niche for the maintenance and repair of skeletal muscle after birth.

Epigenetic modifications refer to chemical modifications that regulate gene expression without altering the DNA sequence. These include DNA methylation, histone modifications, and regulation by non-coding RNAs (ncRNAs). 5mC and 5hmC are distinct types of DNA modifications that play crucial roles in epigenetic regulation. 5mC is concentrated in promoter regions and performs extremely important functions in gene silencing, genomic imprinting, suppressing transposable elements, and repetitive sequences, relating to the inhibition of gene expression. Conversely, 5hmC counteracts these effects by being concentrated in specific genomic regions, regulating gene activation, and preventing excessive gene silencing due to abnormal DNA methylation. Both modifications interact to maintain the dynamic balance between gene activation and silencing [24]. In conjunction with our research, we observed dynamic changes in the modification level of DNA demethylation-modified 5mC in muscle satellite cells. The modification level of 5hmC modified by demethylation also showed dynamic changes, but the trends in 5mC and 5hmC in yak muscle satellite cells were not entirely opposite. This was particularly evident in the three developmental stages: muscle length 17–21 mm, 98–12 mm, and 250–610 mm. In mammals, regions of DNA methylation and demethylation are often associated with tissue-specific gene expression. Ponnaluri VK [25] suggested that methylation (5mC) in myogenic cells and muscle fibers may be linked to demethylation processes in skeletal muscle, influencing gene regulation. Moreover, Shi K [26] found that the Tet2 gene facilitates the differentiation of avian myogenic cells by promoting DNA demethylation. Knocking down the Tet2 gene led to a significant decrease in 5hmC levels, further highlighting the critical role of DNA demethylation in muscle development. The primary myofibers of yak skeletal muscle began to form at the fetal stage of 21 mm (as shown in Section 3.1). It is preliminarily inferred that the gradual decrease in the ratio of PAX7 + 5mC^+^/PAX7^+^ is associated with cell differentiation, and the modification level of 5mC gradually decreased during the differentiation of muscle satellite cells into myotubes. It is preliminarily inferred that the interplay between 5mC and 5hmC plays a critical role in modulating the expression of genes crucial for muscle differentiation, with 5mC typically repressing gene activity and 5hmC facilitating gene activation. The transition to a muscle length of 98 mm represents a critical stage in muscle development, coinciding with the full formation of primary myofibers. Therefore, we hypothesize that the dynamic fluctuations of 5mC and 5hmC in muscle satellite cells may be linked to the proliferation and differentiation processes occurring after the formation of primary myofibers. As satellite cells differentiate into muscle fibers, their epigenetic landscape—particularly DNA methylation and demethylation—undergoes dynamic changes that regulate gene expression and contribute to muscle fiber formation.

H3K4me3 is one of the most extensively studied post-translational modifications of histones, and its trimethylation is usually associated with active gene expression. Some researchers have found that MLL1 mediates the epigenetic regulation of Myf5 through H3K4me3 at the promoter, thereby promoting the proliferation of myoblasts and MuSCs [27,28]. In this study, when fetal bovines develop to around 11 weeks to 110 days, almost all MuSCs are modified by H3K4me3. While H3K4me3 is generally associated with active transcription and cell proliferation, our findings suggest that the proliferation of satellite cells slows down around 70 days, potentially reflecting changes in the satellite cell population or the initiation of differentiation.

The modification levels of 5mC, 5hmC, and H3K4me3 do not directly correlate with gene expression levels. The location of these modifications, however, is crucial for linking them to gene activity and different developmental stages. At different stages of embryonic development, gene expression patterns change dramatically, with epigenetic modifications playing a pivotal role in regulating these transitions. During the early stages of fetal development, certain genes need to be tightly suppressed to ensure normal morphogenesis. This repression is often mediated by 5mC modifications in key regulatory regions, such as promoters, preventing the premature activation of these genes. As cells differentiate, previously repressed genes need to be activated to promote differentiation in a specific direction. This activation is often associated with a gradual decrease in 5mC modifications in the promoter regions of these genes. At the same time, activation-related modifications such as 5hmC or H3K4me3 accumulate at the promoter or enhancer regions of these genes, facilitating their activation. This leads to a shift in gene expression patterns that supports the developmental needs of the cell. Various types of ncRNAs, such as microRNAs and long non-coding RNAs, account for a significant portion of total RNA in cells and play crucial roles in regulating myogenesis. These ncRNAs modulate gene expression by interacting with chromatin, RNA transcripts, and regulatory proteins, thereby influencing muscle cell differentiation [20]. In addition, we can further explore the effects of chromatin structure recombination and ncRNAs on yak muscle satellite cells during prenatal development of yak skeletal muscle by combining the comprehensive three-dimensional structural characterization of genomes and transcriptomics from the primary myofiber formation stage to the secondary myofiber formation stage.

## 5. Conclusions

This study aims to explore the developmental characteristics of MuSCs during yak fetal development and their relationship with epigenetic modifications. Using H&E staining and immunofluorescence techniques to analyze the longissimus dorsi of yak fetuses at different developmental stages, key time points for skeletal muscle growth and development in yak fetuses were identified. Additionally, the study found that DNA methylation and histone modifications are closely related to skeletal muscle growth and development during the fetal period. These findings provide important references and theoretical foundations for further research into the molecular mechanisms of muscle growth and development in yaks.

## Figures and Tables

**Figure 1 biology-13-01091-f001:**
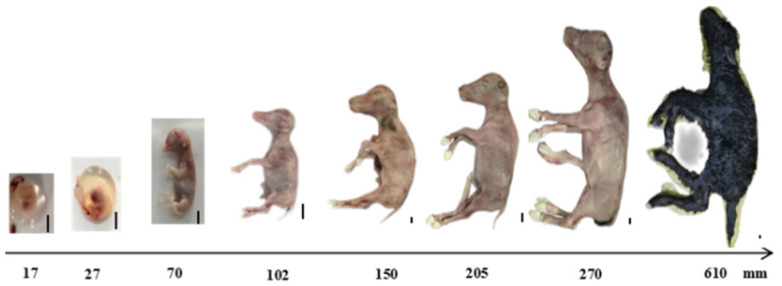
Different development lengths of fetal calves pictures; bar = 10 mm. Images of 8 fetal calves were selected, with lengths of 17, 27, 70, 102, 150, 205, 270, and 610 mm. With increasing length, the morphology of the fetal calves changes until at birth, the entire body is covered with hair.

**Figure 2 biology-13-01091-f002:**
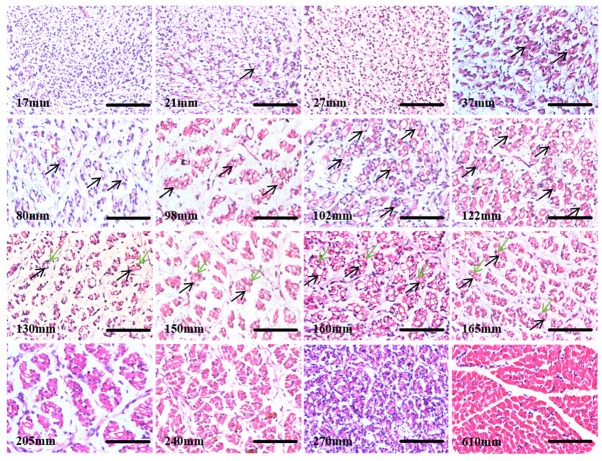
Yak fetal calf different development stages back longest muscle tissue HE staining; bar = 50 μm; *n* = 3. Black arrows, primary muscle fibers; green arrows, secondary muscle fibers. With the continuous increase of calf length, muscle fibers gradually increase and become complete, and just before birth, muscle fibers are neatly arranged into muscle bundles.

**Figure 3 biology-13-01091-f003:**
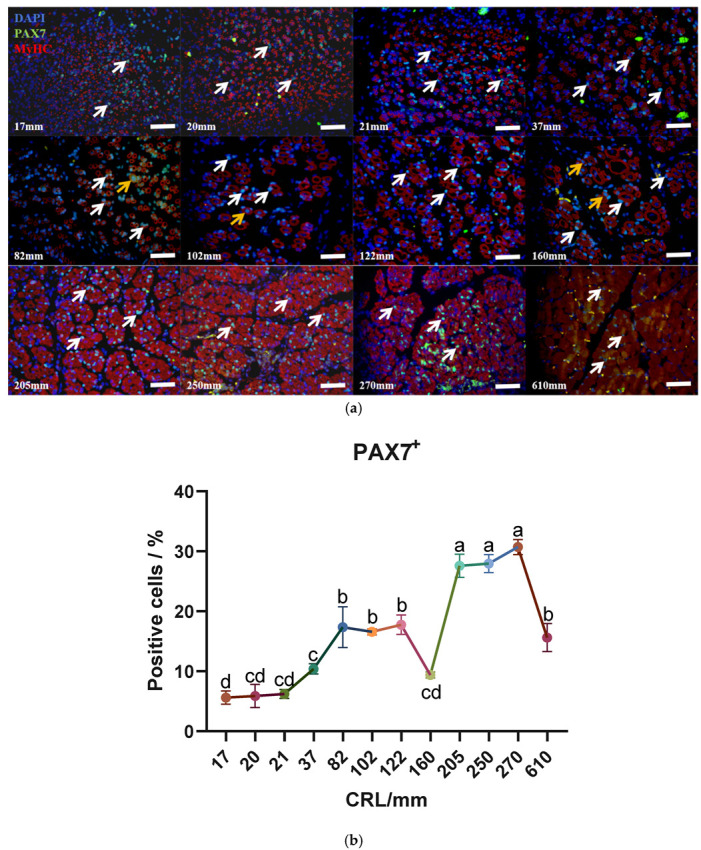
Localization of PAX7^+^ cells and MYHC at different developmental lengths and quantification of PAX7^+^ cells. (**a**) Localization of PAX7^+^ cells and MYHC at different developmental lengths. Bar = 50 μm; *n* = 3. White arrows: PAX7^+^ cells; orange arrows: PAX7^+^ cells located in the middle of the myotube. (**b**) Number of PAX7^+^ cells in cross-section of the longissimus dorsi muscle at gestational age. Data are presented as the mean ± SEM of three fetal cattle, with three longissimus dorsi cross-sections analyzed per individual. Different letters indicate significant differences between groups (*p* < 0.05). Total cellular score: 17 mm, *n* = 939; 20 mm, *n* = 1055; 21 mm, *n* = 753; 37 mm, *n* = 1105; 82 mm, *n* = 1115; 102 mm, *n* = 1149; 122 mm, *n* = 791; 160 mm, *n* = 1195; 205 mm, *n* = 707; 250 mm, *n* = 1395; 270 mm, *n* = 744; 610 mm, *n* = 807.

**Figure 4 biology-13-01091-f004:**
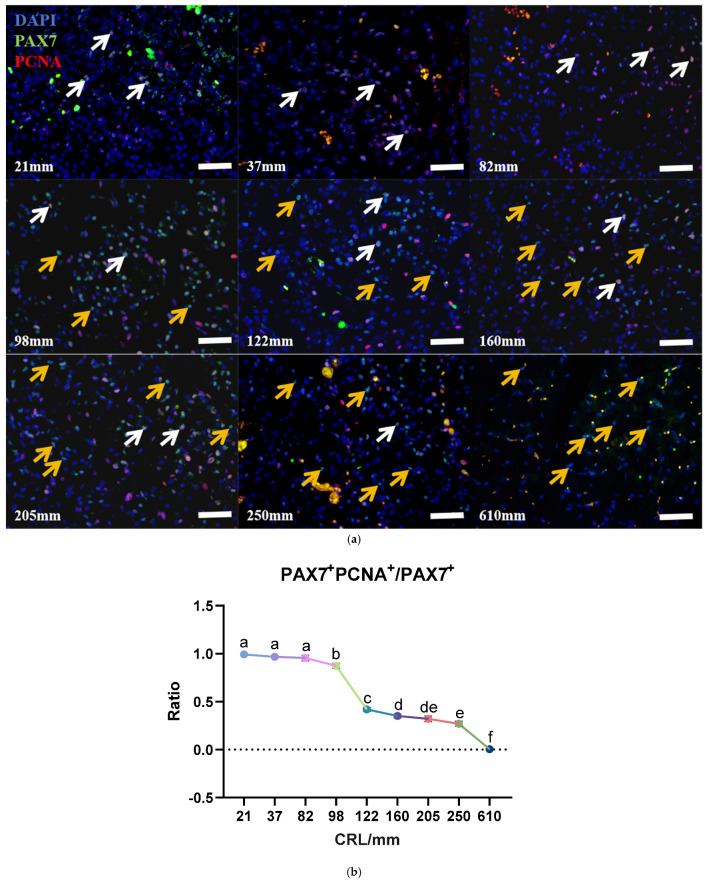
Proliferation of PAX7^+^ cells with different developmental lengths. (**a**) PAX7 and PCNA immunofluorescence co−staining with different developmental lengths; bar = 50 μm; *n* = 3. White arrows: PAX7^+^/PCNA^+^ cells; orange arrows: PAX7^+^/PCNA^−^ cells. (**b**) Ratio of PAX7^+^PCNA^+^/PAX7^+^ cells. Data are presented as the mean ± SEM of three fetal cattle, with three longissimus dorsi cross-sections analyzed per individual. Different letters indicate significant differences between groups (*p* < 0.05). Total cellular score: 21 mm, *n* = 2931; 37 mm, *n* = 1812; 82 mm, *n* = 1881; 122 mm, *n* = 1875; 160 mm, *n* = 1773; 205 mm, *n* = 1674; 250 mm, *n* = 1800; 610 mm, *n* = 1710.

**Figure 5 biology-13-01091-f005:**
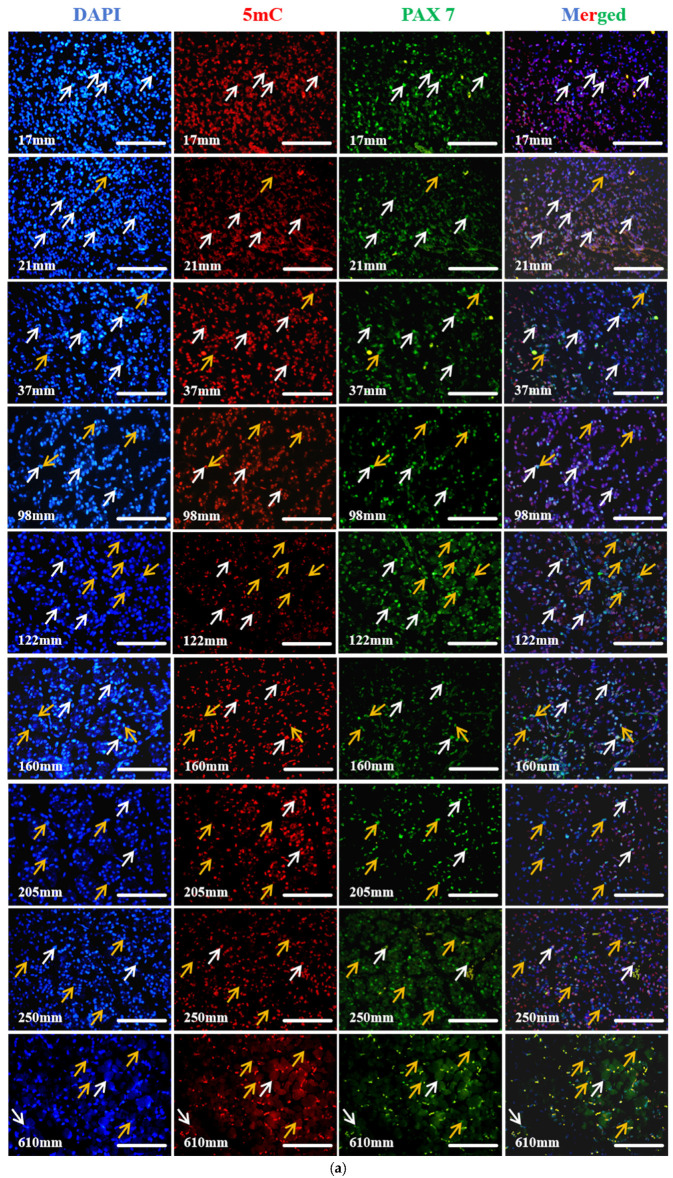
MuSCs and DNA methylation studies at different developmental lengths. (**a**) Co-staining results of PAX7 and 5mC at different developmental lengths. White arrows: PAX7^+^/5mC^+^ cells; orange arrows: PAX7^+^/5mC^−^ cells; bar = 50 μm; *n* = 3. (**b**) Ratio of PAX7^+^5mC^+^/PAX7^+^ cells. Data are presented as the mean ± SEM of three fetal cattle, with three longissimus dorsi cross-sections analyze per individual. Different letters indicated significant differences between groups (*p* < 0.05). PAX7^+^/5mC^+^ cells: 17 mm, *n* = 106; 21 mm, *n* = 354; 37 mm, *n* = 114; 98 mm, *n* = 174; 122 mm, *n* = 48; 160 mm, *n* = 90; 205 mm, *n* = 42; 250 mm, *n* = 144; 610 mm, *n* = 228. (**c**) Co-staining results of PAX7 and 5hmC with different developmental lengths. White arrows: PAX7^+^/5hmC^+^ cells; orange arrows: PAX7^+^/5hmC^−^ cells; bar = 50 μm; *n* = 3. (**d**) Ratio of PAX7^+^5 mC^+^/PAX7^+^ cells. Data are presented as the mean ± SEM of three fetal cattle, with three longissimus dorsi cross-sections analyzed per individual. Different letters indicate significant differences between groups (*p* < 0.05). PAX7^+^/5hmC^+^ cells: 17 mm, *n* = 96; 21 mm, *n* = 48; 37 mm, *n* = 132; 98 mm, *n* = 108; 122 mm, *n* = 90; 160 mm, *n* = 144; 205 mm, *n* = 78; 250 mm, *n* = 102; 610 mm, *n* = 216.

**Figure 6 biology-13-01091-f006:**
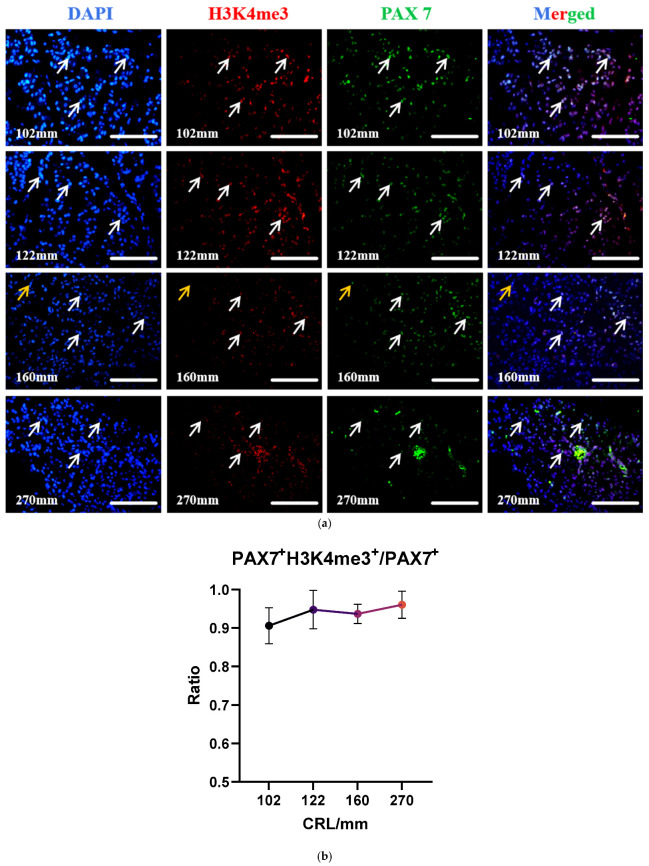
Expression of H3K4me3 in MuSCs with different developmental lengths. (**a**) Co-staining results of PAX7 and H3K4me3 with different developmental lengths. White arrows: PAX7^+^/H3K4me3^+^ cells; orange arrows: PAX7^+^/H3K4me3^−^ cells; bar = 50 μm; *n* = 3. (**b**) Ratio of PAX7+H3K4me3^+^/PAX7^+^ cells. Data are presented as the mean ±SEM of three fetal cattle, with three longissimus dorsi cross-sections analyzed per individual. PAX7^+^/H3K4me3^+^ cells: 102 mm, *n* = 264; 122 mm, *n* = 150; 160 mm, *n* = 174; 270 mm, *n* = 474.

**Table 1 biology-13-01091-t001:** Fetal calf CRL and GA.

Number	1	2	3	4	5	6	7	8	9	10	11	12	13	14	15	16	17	18
CRL/mm	17	21	27	37	70	80	98	102	122	130	150	160	165	205	240	250	270	610
GA/d	37	41	45	48	62	69	76	78	89	90	100	104	110	125–135			140–147	240–245
Sample Size	3	3	3	3	3	4	3	3	3	5	4	3	3	6	3	3	3	3

## Data Availability

All data generated or analyzed during this study are included in this published article.

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
