# Peer review of "The Development of Prenatal Muscle Satellite Cells (MuSCs) and Their Epigenetic Modifications During Skeletal Muscle Development in Yak Fetus"

_biology, 2024, doi:10.3390/biology13121091_

Round 1

Reviewer 1 Report

Comments and Suggestions for Authors

SummaryThe study creatively explores the properties of yak muscle development during the fetal stage. It reveals specific morphological characteristics associated with the growth of the fetus. Additionally, the authors utilized immunofluorescent staining to identify the developmental profile of satellite cells. To further expand the understanding of fetal myogenesis in yaks, the study incorporates an analysis of dynamic epigenetic modifications. Collectively, the research provides a relatively comprehensive multidimensional atlas of prenatal yak muscle development.

1.     Table 1: The table should be split into two separate tables for clarity and better presentation. 

2.     The manuscript should specify the number of animals used at each time point. 

3.     The background of the immunofluorescent images is not clean. For instance, in Figure 3, PAX7 stain (green) is expected to be in the nuclei, but your figure shows that it is also in the cytoplasm. Similarly, in another image, PCNA stains the cytoplasm red. The manuscript should provide high-quality images with clearer and more accurate staining method. 

4.     Due to the small and highly condensed nature of satellite cells, staining the nucleus with DAPI is necessary to distinguish satellite cells from non-specific staining. Additionally, immunofluorescent images should be presented in separate channels to highlight specific signals. For instance, in Figure 5, the signal marked by the orange arrow appears to be non-specific. 

5.     Figure 1.2: The resolution of the section images appears inconsistent, with variations that suggest they may not have been captured at the same magnification. For clarity and comparability, it is recommended that all images be obtained at identical magnifications, regardless of the scale bar calibration. This discrepancy is particularly noticeable in the bottom-left image (205mm). 

6.     Although using crown-rump length as an index to describe developmental stages is reasonable in certain contexts, both gestational age and crown-rump length are used in this manuscript to describe developmental events, leading to confusion between the two systems. Adding a line to clarify the relationship between crown-rump length, gestational age, and developmental stage could help readers better understand how these measures are related. 

7.     5mC and 5hmC should be quantified using ELISA or other methods. Immunofluorescent staining alone cannot provide conclusive evidence that 5mC levels fluctuate during muscle development (as mentioned in abstract lines 18-20 and the subsequent main paragraph). 

8.     The modification levels of 5mC, 5hmC, and H3K4me3 cannot directly indicate gene expression levels. The location of modifications is more critical for linking epigenetic modifications to gene expression and developmental stages.  

9.     The study should include statistical analysis to validate the conclusions. Dynamic changes cannot be validated solely by immunofluorescent images without statistical support. Cell counting is necessary: counting cells with or without specific gene expression, calculating the ratio between them, and generating the dynamic change in these ratios over time will strengthen the conclusions.
For example, in sections 3.2 (the ratio of PAX7+/total cell number) and 3.3 ((PAX7+/PCNA+)/(PAX7+/PCNA-)), these are just two instances; many other statistical analyses should also be conducted. 

Comments on the Quality of English Language

1.     The full term should be provided for every abbreviation upon its first occurrence.

2.     Line 332: The abbreviation "DMR" for "regions of methylation or demethylation" is misleading, as the first letters of the words do not correspond to "DMR." Additionally, in DNA methylation studies, "DMR" is commonly used to denote "Differentially Methylated Region," which may cause confusion. Using a different abbreviation would help clarify the term and prevent potential misunderstanding. 

3.     The manuscript should adopt professional terminology, and the term "muscle tube" should be replaced with a more appropriate expression. 

4.     Line 119: The title of the book should be formatted in italic font.

5.     The writing should be thoroughly revised and significantly improved, and the paragraphs should be reorganized to enhance clarity. The logical gaps hinder a full understanding of the content. 

Author Response

Response to Reviewer 1 Comments

1. Summary

2. Questions for General Evaluation

Reviewer’s Evaluation

Response and Revisions

Does the introduction provide sufficient background and include all relevant references?

Yes

Are all the cited references relevant to the research?

Yes

Is the research design appropriate?

Yes

Are the methods adequately described?

Yes

Are the results clearly presented?

Yes

Are the conclusions supported by the results?

Yes

3. Point-by-point response to Comments and Suggestions for Authors

Comments 1: Table 1: The table should be split into two separate tables for clarity and better presentation. 

Response 1:Thank you for pointing this out. We agree with this comment. For the sake of understanding, we have supplemented Table 1 and divided it into two tables.

Comments 2: The manuscript should specify the number of animals used at each time point. 

Response 2: Agree. We have supplemented the sample size in Table 1.

Comments 3: The background of the immunofluorescent images is not clean. For instance, in Figure 3, PAX7 stain (green) is expected to be in the nuclei, but your figure shows that it is also in the cytoplasm. Similarly, in another image, PCNA stains the cytoplasm red. The manuscript should provide high-quality images with clearer and more accurate staining method. 

Response 3: Thank you for pointing this out. We agree with this comment. Therefore, We have done our best to re-process the background of the immunofluorescence image, and to make it easier to confirm that PAX7 stain is located on the nucleus, we have fully demonstrated the staining of the nuclei by DAPI on Figure 3.

Comments 4: Due to the small and highly condensed nature of satellite cells, staining the nucleus with DAPI is necessary to distinguish satellite cells from non-specific staining. Additionally, immunofluorescent images should be presented in separate channels to highlight specific signals. For instance, in Figure 5, the signal marked by the orange arrow appears to be non-specific. 

Response 4: We agree with this comment. Therefore, in order to distinguish between satellite cells and non-specific staining, we have all added DAPI staining in Figures 3, 4, and 5, and have presented specific stains as separate channels in Figures 4 and 5. In addition, we further checked the arrows of all the annotations in the figure.

Comments 5: Figure 1.2: The resolution of the section images appears inconsistent, with variations that suggest they may not have been captured at the same magnification. For clarity and comparability, it is recommended that all images be obtained at identical magnifications, regardless of the scale bar calibration. This discrepancy is particularly noticeable in the bottom-left image (205mm). 

Response 5: Thank you very much for your comments, we double-checked the clarity and magnification of all the pictures in Figure 1.2, and for the lower left corner of the picture (205mm) you mentioned, it was found that its magnification is correct. In order to present the HE staining results more accurately and clearly, we have rearranged the images and added a 240mm result map.

Comments 6: Although using crown-rump length as an index to describe developmental stages is reasonable in certain contexts, both gestational age and crown-rump length are used in this manuscript to describe developmental events, leading to confusion between the two systems. Adding a line to clarify the relationship between crown-rump length, gestational age, and developmental stage could help readers better understand how these measures are related. 

Response 6:Thank you for pointing this out. We agree with this comment. Therefore, We have added "CRL is an important parameter that is closely associated with GA and developmental stage. Generally, as gestational age progresses, the crown - rump length increases. In the early stages of pregnancy, specific crown - rump length values correspond to particular developmental milestones. As the weeks pass, the crown - rump length lengthens, and the embryo develops more complex organs and systems. The developmental stage can be estimated based on the crown - rump length and gestational age, with each stage having characteristic morphological and physiological features. " to Outcome 3.1.1 to elicit the relationship between Crown-Rump Length, gestational age, and developmental stage.

Comments 7: 5mC and 5hmC should be quantified using ELISA or other methods. Immunofluorescent staining alone cannot provide conclusive evidence that 5mC levels fluctuate during muscle development (as mentioned in abstract lines 18-20 and the subsequent main paragraph).

Response 7: Thank you for pointing this out. We mainly focused on the methylation and demethylation status of satellite cells, without paying attention to the methylation level in the entire muscle tissue. Moreover, through supplementary statistical analysis (as shown in Figures 4c and 4d), we did observe some cells with low methylation levels in satellite cells, indicating that the development or differentiation of muscle satellite cells during development is related to methylation or demethylation levels.

Comments 8: The modification levels of 5mC, 5hmC, and H3K4me3 cannot directly indicate gene expression levels. The location of modifications is more critical for linking epigenetic modifications to gene expression and developmental stages.  

Response 8: Thank you for pointing this out. The modification levels of 5mC, 5hmC, and H3K4me3 are not directly equivalent to gene expression levels by themselves, and the location of these modifications is the key factor in accurately linking epigenetic modifications to gene expression and different developmental stages. This is also an important follow-up to this study, which uses bisulfite sequencing, ChIP-Chip, and ChIP-Seq to determine the relationship between epigenetic modifications and gene expression. In this study, we are only looking at the possibility that PAX7+ cells may be affected by these epigenetic modifications from the perspective of immunofluorescence staining. So we added reflection on that in the discussion section.”The modification levels of 5mC, 5hmC, and H3K4me3 are not directly equivalent to gene expression levels by themselves, and the location of these modifications is the key factor in accurately linking epigenetic modifications to gene expression and different developmental stages. At different stages of embryonic development, the pattern of gene expression changes dramatically. Positional changes in epigenetic modifications play a key role in this. During the early stages of embryonic development, when certain genes need to be severely suppressed to ensure normal morphogenesis, a large number of 5mC modifications may occur in key regulatory regions of these genes, such as promoters, to ensure that these genes are not incorrectly activated. As cells differentiate, some previously repressed genes need to be activated to promote cell differentiation in a specific direction, and 5mC modifications may gradually decrease in the promoter regions of these genes, while activation-related modifications such as 5hmC or H3K4me3 appear or increase at appropriate locations (e.g., near enhancers, transcription start sites), so as to achieve a shift in gene expression patterns to accommodate developmental needs.”

Comments 9: The study should include statistical analysis to validate the conclusions. Dynamic changes cannot be validated solely by immunofluorescent images without statistical support. Cell counting is necessary: counting cells with or without specific gene expression, calculating the ratio between them, and generating the dynamic change in these ratios over time will strengthen the conclusions.
For example, in sections 3.2 (the ratio of PAX7+/total cell number) and 3.3 ((PAX7+/PCNA+)/(PAX7+/PCNA-)), these are just two instances; many other statistical analyses should also be conducted. 

Response 9: Thank you for pointing this out. We have added statistical analysis charts on cell ratios, including Figures 2b, 3b, 4b, 4d, and 5b.

4. Response to Comments on the Quality of English Language

Point 1: The full term should be provided for every abbreviation upon its first occurrence.

Response 1: Thank you for pointing this out. We have carefully read and revised the article, and have changed the abbreviation of technical terms, for example, Crown-Rump Length (CRL)、Muscle satellite cell (MuSCs) and Gestational Age (GA) et all.

Point 2: Line 332: The abbreviation "DMR" for "regions of methylation or demethylation" is misleading, as the first letters of the words do not correspond to "DMR." Additionally, in DNA methylation studies, "DMR" is commonly used to denote "Differentially Methylated Region," which may cause confusion. Using a different abbreviation would help clarify the term and prevent potential misunderstanding. 

Response 2Thank you very much for pointing this out, we have removed the abbreviation "DMR" in the discussion section.

Point 3: The manuscript should adopt professional terminology, and the term "muscle tube" should be replaced with a more appropriate expression. 

Response 3: Thank you for pointing this out. We have changed the "muscle tube" in three places in the article to " Myotube ".

Point 4:  Line 119: The title of the book should be formatted in italic font.

Response 4: Agree. We have booked Veterinary Obstetrics in italics,”Veterinary Obstetrics”.

Point 5: The writing should be thoroughly revised and significantly improved, and the paragraphs should be reorganized to enhance clarity. The logical gaps hinder a full understanding of the content. 

Response 5: Thank you for pointing this out. We have thoroughly revised the manuscript under the guidance of professional teachers, and all the red parts in the manuscript are the revised parts. We warmly welcome any further questions you may have regarding the new manuscript. Thank you again for your time and consideration.

Reviewer 2 Report

Comments and Suggestions for Authors

1. Introduction

1.1 Although this section provides detailed information on muscle fiber development, the continuity and articulation of the different stages of skeletal muscle development (e.g., from primary to secondary muscle canals) are not clearly expressed, which may cause some confusion for readers to understand the muscle fiber development process. It is suggested to optimize according to the idea of "formation and development of primary muscle ducts - formation and development of secondary muscle ducts - development of muscle fibers".

1.2 The expression of the sentence "Nonetheless, understanding of how epigenetic modifications regulate skeletal muscle cell proliferation and differentiation during yak fetal development remains limited" is very general and vague. What specific aspects of the research are still unclear, and can this study fill these gaps to some extent?

1.3 Please add how this study will improve the production performance of other livestock and poultry and what positive effects it will have on the development of herbivorous animal husbandry, which will help to enhance the industrial significance and scientific value of this study.

Discussion

2.1 The statement "the timing of primary and secondary muscle fiber formation differs between yaks and beef cattle, which may be attributed to species differences" can be discussed to provide more details on how these differences affect yak muscle growth or quality. For example, how development time affects muscle structure, differences in muscle development, quality, and flavor between yaks and beef cattle, and briefly explain why these differences are important for yak production.

2.2 The statement "this aligns with our experimental results" is not clear enough. When citing the work of others (e.g., Kuang S, Rocheteau P), authors should specify how the results of this study confirm or challenge previous research.

2.3 As satellite cells differentiate into muscle fibers, their epigenetic landscape, particularly DNA methylation and demethylation, undergoes dynamic changes that regulate gene expression and promote muscle fiber formation. This point is not clearly expressed in the text, it is suggested to modify the relevant expression.

2.4 Are you suggesting that the reduction in 5mC is a direct driver of differentiation?  Or is it a consequence of other factors, such as signaling pathways? Further complementation of how the 5mC/5hmC balance affects the expression of muscle-specific genes may give readers a better understanding.

2.5 The statement that H3K4me3 "usually promotes gene transcription and enhances satellite cell proliferation" could be slightly confusing when juxtaposed with the observed slowdown in proliferation.   Consider rephrasing it to: "While H3K4me3 is generally associated with active transcription and cell proliferation, our findings suggest that the proliferation of satellite cells slows down around 70 days, potentially reflecting changes in the satellite cell population or the initiation of differentiation."

Comments on the Quality of English Language

This manuscript needs editing by experts with expertise in technical English editing paying particular attention to English grammar, spelling, and sentence structure, so that the goals and results of the study are clear to the potential readers. Of course, if the author used AI software for language editing, this needs to be declared in the manuscript.

Author Response

Response to Reviewer 2 Comments

1. Summary

2. Questions for General Evaluation

Reviewer’s Evaluation

Response and Revisions

Does the introduction provide sufficient background and include all relevant references?

Yes

Are all the cited references relevant to the research?

Yes

Is the research design appropriate?

Yes

Are the methods adequately described?

Yes

Are the results clearly presented?

Yes

Are the conclusions supported by the results?

Yes

3. Point-by-point response to Comments and Suggestions for Authors

Comments 1: Although this section provides detailed information on muscle fiber development, the continuity and articulation of the different stages of skeletal muscle development (e.g., from primary to secondary muscle canals) are not clearly expressed, which may cause some confusion for readers to understand the muscle fiber development process. It is suggested to optimize according to the idea of "formation and development of primary muscle ducts - formation and development of secondary muscle ducts - development of muscle fibers".

Response 1:Thank you for pointing this out. We agree with this comment. Therefore, we have added a sentence to the third paragraph of the introduction to help the reader understand more clearly the progress of muscle fiber development. "The process of skeletal muscle development is sequential, starting with the formation of primary myotubes and progressing to secondary myotubes, which form in greater numbers as development advances."

Comments 2: The expression of the sentence "Nonetheless, understanding of how epigenetic modifications regulate skeletal muscle cell proliferation and differentiation during yak fetal development remains limited" is very general and vague. What specific aspects of the research are still unclear, and can this study fill these gaps to some extent?

Response 2: Agree. Based on current research, there is still relatively little research on how DNA methylation and histone modification specifically affect the muscle development of yak fetuses. This study can fill the gap in this area. Therefore, we will change ”Nonetheless, understanding of how epigenetic modifications regulate skeletal muscle cell proliferation and differentiation during yak fetal development remains limited.” to “However, despite the growing interest in epigenetics, little is known about how DNA methylation and histone modification specifically influence yak fetal muscle development, particularly in the context of satellite cells."

Comments 3: Please add how this study will improve the production performance of other livestock and poultry and what positive effects it will have on the development of herbivorous animal husbandry, which will help to enhance the industrial significance and scientific value of this study.

Response 3: We agree with this comment. Therefore, We have revised the last sentence of the introduction to “This research aims to explore the changes in satellite cells and their associated epigenetic modifications during yak fetal development, with a particular focus on DNA methylation and histone modifications. The findings may provide new insights into yak skeletal muscle growth and development, which could inform strategies to enhance production performance in the future.”

Comments 4: The statement "the timing of primary and secondary muscle fiber formation differs between yaks and beef cattle, which may be attributed to species differences" can be discussed to provide more details on how these differences affect yak muscle growth or quality. For example, how development time affects muscle structure, differences in muscle development, quality, and flavor between yaks and beef cattle, and briefly explain why these differences are important for yak production.

Response 4:We agree with this comment. Therefore, We have revised the discussion section from "the timing of primary and secondary muscle fiber formation differs between yaks and beef cattle, which may be attributed to species differences" to “These differences in muscle fiber development timing may contribute to the unique texture and quality of yak meat, which is known for its distinctive characteristics compared to beef.”

Comments 5:The statement "this aligns with our experimental results" is not clear enough. When citing the work of others (e.g., Kuang S, Rocheteau P), authors should specify how the results of this study confirm or challenge previous research.

Response 5: Agree. When citing the study Kuang S, we have changed “this aligns with our experimental results” to "Our results support Kuang’s findings on the asymmetrical division of satellite cells and suggest that the differentiation of PAX7+/Myf5+ satellite cells at critical developmental stages plays a key role in the formation of new muscle fibers."

Comments 6:As satellite cells differentiate into muscle fibers, their epigenetic landscape, particularly DNA methylation and demethylation, undergoes dynamic changes that regulate gene expression and promote muscle fiber formation. This point is not clearly expressed in the text, it is suggested to modify the relevant expression.

Response 6:Thank you for pointing this out. We agree with this comment. Therefore, We added "As satellite cells differentiate into muscle fibers, their epigenetic landscape, particularly DNA methylation and demethylation, undergoes dynamic changes that regulate gene expression and contribute to muscle fiber formation." at the end of the third paragraph of our discussion.

Comments 7:Are you suggesting that the reduction in 5mC is a direct driver of differentiation?  Or is it a consequence of other factors, such as signaling pathways? Further complementation of how the 5mC/5hmC balance affects the expression of muscle-specific genes may give readers a better understanding.

Response 7: The interaction between 5mC and 5hmC plays a role in certain genes involved in muscle differentiation. Therefore, we will modify the third paragraph of the discussion section from "the gradual reduction of PAX7+/5mC+ cells could be associated with cellular differentiation" to "The interplay between 5mC and 5hmC helps to finely tune the expression of genes crucial for muscle differentiation, with 5mC typically repressing gene activity and 5hmC facilitating gene activation.".

Comments 8:The statement that H3K4me3 "usually promotes gene transcription and enhances satellite cell proliferation" could be slightly confusing when juxtaposed with the observed slowdown in proliferation.   Consider rephrasing it to: "While H3K4me3 is generally associated with active transcription and cell proliferation, our findings suggest that the proliferation of satellite cells slows down around 70 days, potentially reflecting changes in the satellite cell population or the initiation of differentiation."

Response 8: Thank you for pointing this out. We agree with this comment. Therefore, we have revised " H3K4me3 is believed to promote the initiation of gene transcription and enhance the proliferation of muscle satellite cells" to "While H3K4me3 is generally associated with active transcription and cell proliferation, our findings suggest that the proliferation of satellite cells slows down around 70 days, potentially reflecting changes in the satellite cell population or the initiation of differentiation." in the last paragraph of the discussion.

4. Response to Comments on the Quality of English Language

Point 1:This manuscript needs editing by experts with expertise in technical English editing paying particular attention to English grammar, spelling, and sentence structure, so that the goals and results of the study are clear to the potential readers. Of course, if the author used AI software for language editing, this needs to be declared in the manuscript.

Response 1:  Thank you very much for your feedback. We have consulted experts in the relevant field to revise the language, grammar, and sentence structure of this manuscript. If you have any further questions, please provide them in detail and we can make specific modifications based on your concerns.

We warmly welcome any further questions you may have regarding the new manuscript. Thank you again for your time and consideration. 

Reviewer 3 Report

Comments and Suggestions for Authors

The core of this paper is to study the developmental characteristics of satellite cells and the dynamic changes of their epigenetic modifications (DNA methylation and histone modification) during skeletal muscle development in yak fetuses. Through morphological observation and immunofluorescence staining techniques, the study focused on analyzing the distribution, proliferation status, and epigenetic characteristics of skeletal muscle satellite cells in different periods of yak fetuses, and revealed the influence of key time points on muscle development. Specifically, the authors found that primary muscle fibers were formed on the 40th day of embryonic development, fully formed at 11 weeks, and secondary muscle fibers were mainly formed at 105 days; satellite cells showed a dynamic proliferation pattern and stopped proliferation at critical time points, and were eventually stored in stem cell banks for use after birth; DNA methylation (5mC) gradually decreased, DNA demethylation (5hmC) showed a dynamic trend of increase, decrease and increase again, while histone H3K4me3 modification played a dominant role in a specific period.

However, there are still some points that deserve improvement in the article.

1. The study is mainly based on observations and changes in epigenetic marks but lacks verification of the specific role of these modifications in gene expression regulation or muscle function.

2. Although it covers multiple embryonic development stages, the number of samples may not be enough to exclude the influence of individual differences, and it does not involve in-depth analysis at the single-cell level.

3. Only DNA methylation and histone H3K4me3 modification were studied, and other important epigenetic regulatory mechanisms such as non-coding RNA or chromatin structure changes were not involved. This at least should be discussed in the Discussion.

Author Response

Response to Reviewer 3 Comments

1. Summary

2. Questions for General Evaluation

Reviewer’s Evaluation

Response and Revisions

Does the introduction provide sufficient background and include all relevant references?

Yes

Are all the cited references relevant to the research?

Yes

Is the research design appropriate?

Yes

Are the methods adequately described?

Yes

Are the results clearly presented?

Yes

Are the conclusions supported by the results?

Yes

3. Point-by-point response to Comments and Suggestions for Authors

Comments 1: The study is mainly based on observations and changes in epigenetic marks but lacks verification of the specific role of these modifications in gene expression regulation or muscle function.

Response 1:Thank you for pointing this out. This study mainly used histological and immunofluorescence staining methods to preliminarily investigate the DNA methylation and histone H3K4me3 changes of muscle satellite cells during muscle development. We have supplemented the ratio data (as shown in Figures 4c and 4d), which can intuitively show that the development of muscle satellite cells is related to these epigenetic modifications. We are currently trying to establish an in vitro yak muscle satellite cell culture system, and we will verify the specific role of gene expression regulation on cells in the future.

Comments 2: Although it covers multiple embryonic development stages, the number of samples may not be enough to exclude the influence of individual differences, and it does not involve in-depth analysis at the single-cell level.

Response 2: Thank you for pointing this out. We have added the sample size in Table 1, with a minimum of three replicates per group to exclude individual differences. But for single-cell analysis, we currently do not have a detailed study, which could become our main direction for future research.

Comments 3: Only DNA methylation and histone H3K4me3 modification were studied, and other important epigenetic regulatory mechanisms such as non-coding RNA or chromatin structure changes were not involved. This at least should be discussed in the Discussion.

Response 3: We agree with this comment. We have added a discussion on the impact of non coding RNA or chromatin structural changes on skeletal muscle development in the last paragraph of the discussion section.”As cells differentiate, some previously repressed genes need to be activated to promote cell differentiation in a specific direction, and 5mC modifications may gradually decrease in the promoter regions of these genes, while activation-related modifications such as 5hmC or H3K4me3 appear or increase at appropriate locations, so as to achieve a shift in gene expression patterns to accommodate developmental needs. Various non-coding RNAs (ncRNAs) account for the majority of total RNA in cells and have an important regulatory role in myogenesis. In addition, we can further explore the effects of chromatin structure recombination and ncRNAs on yak muscle satellite cells during prenatal development of yak skeletal muscle by combining the comprehensive three-dimensional structural characterization of genome and transcriptomics from the primary myofiber formation stage to the secondary myofiber formation stage.” 

We warmly welcome any further questions you may have regarding the new manuscript. Thank you again for your time and consideration

Round 2

Reviewer 1 Report

Comments and Suggestions for Authors

The author has made thoughtful revisions to the manuscript and has successfully addressed most of the previously raised concerns. The line charts effectively convey the results, and the new immunofluorescent staining images are well-presented and clear. However, there are still a few issues that may lead to potential confusion for readers. 

Comments on the Quality of English Language

1       The language in lines 481–487 is difficult to comprehend and requires significant revision for clarity. Additionally, the phrase 'expression of 5mC' is not accurate, as 5mC represents a covalent modification of cytosine and is not related to gene expression. It would be more appropriate to describe it as the 'modification level' or 'methylation level' of 5mC. 

2       In line 489, the statement that 'methylation (5mC) in myogenic cells and muscle fibers may be linked to demethylation in skeletal muscle' is unclear and difficult to understand.  

Author Response

Response to Reviewer 1 Comments

1. Summary

2. Questions for General Evaluation

Reviewer’s Evaluation

Response and Revisions

Does the introduction provide sufficient background and include all relevant references?

Yes

Are all the cited references relevant to the research?

Yes

Is the research design appropriate?

Yes

Are the methods adequately described?

Yes

Are the results clearly presented?

Yes

Are the conclusions supported by the results?

Yes

3. Point-by-point response to Comments and Suggestions for Authors

Comments 1: The author has made thoughtful revisions to the manuscript and has successfully addressed most of the previously raised concerns. The line charts effectively convey the results, and the new immunofluorescent staining images are well-presented and clear. However, there are still a few issues that may lead to potential confusion for readers. 

Response 1:Thank you for pointing this out. We agree with this comment. We have read and revised the manuscript many times with the help of experts, and the red part of the manuscript is all the changes that have been made, and we have done our best to complete it.

4. Response to Comments on the Quality of English Language

Point 1: The language in lines 481–487 is difficult to comprehend and requires significant revision for clarity. Additionally, the phrase 'expression of 5mC' is not accurate, as 5mC represents a covalent modification of cytosine and is not related to gene expression. It would be more appropriate to describe it as the 'modification level' or 'methylation level' of 5mC.

Response 1: Thank you for pointing this out. We have changed “expression of 5mC” to “modification level of 5mC”.In addition, we have also changed the expression of the sentence you mentioned.”Epigenetic modifications refer to chemical modifications that regulate gene expression without altering the DNA sequence. These include DNA methylation, histone modifications, and regulation by non-coding RNAs (ncRNAs). ”we observed dynamic changes in the modification level of DNA demethylation-modified 5mC in muscle satellite cells. The modification level of 5hmC modified by demethylation also showed dynamic changes, but the trends in 5mC and 5hmC in yak muscle satellite cells were not entirely opposite. This was particularly evident in the three developmental stages: muscle length 17–21 mm, 98–12 mm, and 250–610 mm. In mammals, regions of DNA methylation and demethylation are often associated with tissue-specific gene expression. ”

Point 2: In line 489, the statement that 'methylation (5mC) in myogenic cells and muscle fibers may be linked to demethylation in skeletal muscle' is unclear and difficult to understand.

Response 2:  Thank you very much for pointing this out, We have modified the expression of the sentence you mentioned.”Ponnaluri VK suggested that methylation (5mC) in myogenic cells and muscle fibers may be linked to demethylation processes in skeletal muscle, influencing gene regulation. Moreover, Shi K found that the Tet2 gene facilitates the differentiation of avian myogenic cells by promoting DNA demethylation. Knocking down the Tet2 gene led to a significant decrease in 5hmC levels, further highlighting the critical role of DNA demethylation in muscle development.”

We warmly welcome any further questions you may have regarding the new manuscript. Thank you again for your time and consideration.
